# ARTIFICIAL PSYCHOLOGY

**Mubarek Mohammed Yesuf,** *
Addis Ababa, Ethiopia
mubarek045@gmail.com,mubarek.mohammed@aait.edu.et

## ABSTRACT

Most Deep Learning based Model Compression methods draw their inspiration from the human brain. This is an example of how powerful but overlooked abstractions from the human mind are. Hoping to benefit the discipline of one from the other, this paper aims to draw attention to a line of research at the intersection of Artificial Intelligence and Psychology, dubbing the area *Artificial Psychology*.

## 1 INTRODUCTION

In vain, many disciplines have tried to define how exactly the human is. On the other hand, the progress in the world of algorithms, particularly Artificial Intelligence algorithms and models, is also growing at an exponential rate. Thousands of research papers push the limits of the state of the art every year. Deliberately or otherwise, there exist certain connections between how the human mind works and concepts from the discipline of Artificial Intelligence and algorithms in general. This paper aims to attract attention towards consolidating and utilizing the techniques and tools invented to study the two worlds separately for solving problems in each.

## 2 MOTIVATION

The fact that a simple perceptron, an abstraction of the human brain in the form of Artificial Neurons, has enabled so much achievement in Artificial Intelligence is overlooked in the research community. Below are some more examples of the intersection of the brain and neural networks, especially concerning: Reinforcement Learning, Neuromorphic computation, etc.

- **Neuromorphic computing** is an area that is getting better attention recently that tries to mimic the computation in the brain into the low-level computation in computers to look more like the ones in the brain where both processing and memory are governed by the neurons and the synapses rather than binary numbers to represent information Schuman et al. (2022).

- **Reinforcement Learning** based algorithms are relatively well studied with respect to human nature, mostly Neuroscience Ludvig et al. (2011), Niv (2009). This is partly because of their intuitive relationship with the concept of stimuli and response being mapped to the concept of action and reward. Beyond that, the explore and exploit trade-off is also well-studied among researchers in Reinforcement Learning.

- **Model compression** is about making deep learning models small while maintaining their performance but it is an area whose connection with the brain is highly overlooked. There are a lot of ways but the main four ones are Pruning, Quantization, Knowledge Distillation, and Low-rank tensor decomposition. Turns out all of these are rooted in how the human mind works: Pruning is related to synaptic pruning, an almost identical concept in neuroscience Paolicelli et al. (2011) that discusses how 'forgetting' keeps us efficient, Quantization is similar to how the human brain encodes and stores information [Gholami et al. (2022),Tee & Taylor (2018)], and low-rank decomposition also has it's in factor analysis, a tool to measure human intelligence Spearman (1961). Knowledge Distillation Hinton et al. (2015) too can be considered as a special form of learning underprivileged information Lopez-Paz et al. (2015) as it implements the teacher-student learning mechanism into an algorithm Vapnik et al. (2015).

---

*

Embedded in these ideas is a good abstraction of the concepts into the world of algorithms. It begs the question if we can find or even design new mappings. Furthermore, it triggers other questions if the other way round could work: whether or not it is possible to use concepts in the space of algorithms in real-world applications.

## 3 ARTIFICIAL PSYCHOLOGY

We are beginning to study the performance of networks as they grow Hestness et al. (2017). But not how they behave while doing that. In humans, this is a psychology study. But it is easier to study that in networks because unlike in humans, in networks, we can also ask the reverse question: how would they behave as they scale down? Furthermore, they are way simpler.

Therefore, this paper is about motivating a line of research that extends the ones mentioned in the previous section dubbing the area *Artificial Psychology* to serve as a common vocabulary for the broad fundamental research questions of *how does a network behave as it goes through changes?* and *Can we design abstractions of concepts in psychology for problems in Artificial Intelligence?*. *Changes* could mean changes in size, capacity, performance, distribution shifts, etc. Specific questions can raise by examining the extensively studied ideas in Psychology such as personality theory Hall et al. (1998), psychoanalysis, etc. A good example of this line of work is Machine Love Lehman (2023) where the author tried to bias a model to be compassionate.

## 4 SIGNIFICANCE

In general, as demonstrated in other cases where two concepts merge such as Neuroscience and Artificial Intelligence, there can be mutually beneficial relationships between the two areas. The following can be mentioned as potential benefits of a study in this line of research.

- Having valuable abstractions at certain levels of the brain and mind, a study in this line of work can give us a more comprehensive view of the **artificially thinking machine** we are trying to build.
- Artificial Intelligence problems such as alignment, explainability, meta-learning, and other behavioral aspects can be studied from a new perspective with the tools and ideas from psychology, and conversely.
- Psychological issues such as personality, self-awareness, psycho-therapy, etc, can be examined from insights gained from Artificial Intelligence.
- This line of study generalizes the study of scaling laws Hestness et al. (2017).
- A new perspective to study emergent behaviors in Large Language models Wei et al. (2022).
- Better affective computing and human-computer interaction.
- We can study Embodied Cognition Varela et al. (2017) for robots, critical for the study of alignment

## 5 CHALLENGES

The above-mentioned concept is required to be substantiated by solid research. To do that there are challenges, and since it requires the merging of two disciplines, the challenges also arise from computational and psychological perspectives. The validity of the idea in light of the Church-Turing thesis is the fundamental challenge. Expertise in only either one of them might not be fruitful: it requires a deeper understanding of both of the subjects.

## 6 CONCLUSION

Out of motivation by the effectiveness of different types of analogies and connections between the human mind and algorithms, a new line of research is suggested at the intersection of Artificial Intelligence and Psychology in an attempt to benefit the study of one from the other.

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
