# OpenReview forum: "ARTIFICIAL PSYCHOLOGY"
_ICLR.cc/2023/TinyPapers — Submitted to Tiny Papers @ ICLR 2023_

### Official Review · Reviewer_piXo · 2023-03-25

**Confidence:** 5

**Summary Of Contributions:**

The paper presents an idea that aims to draw a comparison/an analogy between some aspects of human psychology and algorithms. It presents a few specific examples to drive this idea home.

**Rating:**

Needs Clarification (NC): a submission which does not meet the reviewing criteria and needs clarification for its described problem or solution

**Strengths And Weaknesses:**

Strengths
1. The idea is interesting and has the potential to spark a good discussion

Weaknesses
1. The organization of the ideas presented can be improved. (see corresponding suggestions point)
2. (Section 6) The paper claims that the ideas are conjectures/opinions as of this moment but the challenge to proving them is near impossible as admitted by the paper itself (psychological and computational ideas cannot be compared)
3. Some sentences could be restructured to convey the point more clearly. Example: "Quantization is similar to how we human brain encode and store information"

**Suggested Changes:**

1. The paper does not define what it means by algorithms. Furthermore, some claims are not justified:
	a. "The three major Model Compression methods are Pruning [sic] Quantization, and Knowledge Distillation". The basis for this claim is not provided.
	b. The next claim that all of these are inspired by natural phenomena is not backed sufficiently: "... and Knowledge Distillation as well"
2. Motivation from LLMs: "our next space of experimentation and computation is in the space of ideas." Can the author(s) clarify what they mean by this? Isn't technically every experiment born out of an idea?
3. Section 7 Meta: The statement is too vague. Can the author(s) provide more context? How does the presented idea relate to "meta approach towards anything"
4. Since the paper is over the 2 page main text limit, consider consolidating some ideas.

---

### Official Review · Reviewer_Fx9H · 2023-03-30

**Confidence:** 3

**Summary Of Contributions:**

 Just like there is a connection between brain neurons and neural networks, this paper proposes a connection between algorithms and personalities. This connection can help potentially benefit the study of the mind from the thriving discipline of algorithms.

**Rating:**

Great Start (GS): a submission which meets some of the reviewing criteria but has room for improvement

**Strengths And Weaknesses:**

Strength:
1. This paper proposes an interesting connection which could further open avenues of research.
2. This paper provide specific example of how introvert and extrovert personality gets mapped to algorithm world.

Weakness:
1. This paper is based on personal observation and lacks evidence  to connect to existing research.
2. Large Language models are mentioned very loosely without any relevant connection.

**Suggested Changes:**

Suggested Changes:
1. Have it under 2 pages to adhere to ICLR guideline.
2. Cite and Connect with Existing research papers on the topic.

---

### Author Response · Authors · 2023-05-31
**opt in**

opt in for archival

---

### Meta-Review · Area_Chair_mwZK · 2023-04-05

**Recommendation:** Invite to revise
**Confidence:** 5

**Metareview:**

The paper proposes an interesting connection between introvert and extrovert personalities and the algorithmic world, which could open up new avenues of research. However, the organization of the ideas presented could be improved, and some sentences could be restructured for clarity.Overall, while the idea is intriguing, more research is needed to support these conjectures.

**Summary:**

This paper suggests a connection between algorithms and personalities, similar to the connection between brain neurons and neural networks. By drawing comparisons between aspects of human psychology and algorithms, it aims to benefit the study of the mind and presents specific examples to support this idea.

**Reason For Not Giving A Higher Recommendation:**

I agree with the reviewers that this paper that in its current form it lacks evidence to support its connection between personal observation and existing research. Some claims, such as the three major model compression methods, are not justified. The motivation from large language models is vague. Section 7 is too broad and needs more context. Consider consolidating some ideas as the paper exceeds the 2 page limit.





**Reason For Not Giving A Lower Recommendation:**

N/A

---

### Decision · Program_Chairs · 2023-04-10

No revision received; not invited to archive